

# Linking influenza epidemic onsets to covariates at different scales using a dynamical model

Marion Roussel[1,2], Dominique Pontier[1,2], Jean-Marie Cohen[3], Bruno Lina[4,5] and David Fouchet[1,2]

[1] Laboratoire de Biométrie et Biologie Evolutive URM5558-CNRS, Université de Lyon, Université Claude Bernard Lyon 1, Villeurbanne, France
[2] Université Claude Bernard Lyon 1, LabEx ECOFECT Ecoevolutionary Dynamics of Infectious Diseases, Lyon, France
[3] OPEN ROME (Organize and Promote Epidemiological Network), Paris, France
[4] Laboratory of Virology, Centre National de Référence des Virus Influenzae, Hospices Civils de Lyon, Lyon, France
[5] Virpath, EA4610, Faculty of Medicine Lyon Est, University Claude Bernard Lyon 1, Lyon, France

Corresponding author
Marion Roussel,
marion.roussel1@gmail.com

## ABSTRACT

**Background:** Evaluating the factors favoring the onset of influenza epidemics is a critical public health issue for surveillance, prevention and control. While past outbreaks provide important insights for understanding epidemic onsets, their statistical analysis is challenging since the impact of a factor can be viewed at different scales. Indeed, the same factor can explain why epidemics are more likely to begin (i) during particular weeks of the year (global scale); (ii) earlier in particular regions (spatial scale) or years (annual scale) than others and (iii) earlier in some years than others within a region (spatiotemporal scale).

**Methods:** Here, we present a statistical approach based on dynamical modeling of infectious diseases to study epidemic onsets. We propose a method to disentangle the role of covariates at different scales and use a permutation procedure to assess their significance. Epidemic data gathered from 18 French regions over six epidemic years were provided by the Regional Influenza Surveillance Group (GROG) sentinel network.

**Results:** Our results failed to highlight a significant impact of mobility flows on epidemic onset dates. Absolute humidity had a significant impact, but only at the spatial scale. No link between demographic covariates and influenza epidemic onset dates could be established.

**Discussion:** Dynamical modeling presents an interesting basis to analyze spatiotemporal variations in the outcome of epidemic onsets and how they are related to various types of covariates. The use of these models is quite complex however, due to their mathematical complexity. Furthermore, because they attempt to integrate migration processes of the virus, such models have to be much more explicit than pure statistical approaches. We discuss the relation of this approach to survival analysis, which present significant differences but may constitute an interesting alternative for non-methodologists.

## INTRODUCTION

Influenza is an infectious disease that causes annual epidemics around the world, inducing morbidity in millions of people and a mortality of hundreds of thousands (*World Health Organization, 2014*). Influenza's ability to generate seasonal epidemics and potentially worldwide pandemics makes influenza studies and surveillance a major challenge for public health (*Simonsen, 1999*). However, the mechanisms of its geographic spread and seasonality remain unclear (*Fuhrmann, 2010*; *Lipsitch & Viboud, 2009*). Improving our understanding of the factors that trigger outbreaks is necessary for earlier detection of seasonal epidemics so that public health can be better prepared and efficient preventive/control strategies can be designed.

From a theoretical point of view, influenza epidemic onsets are driven by two phenomena. First, important external flows of infected individuals can help reach a critical number of infected people. Second, local transmission conditions, such as a favorable climate and/or a high density of susceptible humans, should be present.

From an empirical point of view, previous studies have highlighted various covariates that may explain timing differences of influenza epidemics between years and areas. Human movement has been suggested to impact influenza spread (*Charaudeau, Pakdaman & Boëlle, 2014*; *Crépey & Barthélemy, 2007*; *Stark et al., 2012*; *Viboud et al., 2006*). Spatial correlation of influenza epidemics has been observed in major countries (USA (*Viboud et al., 2006*), Canada (*He et al., 2013*; *Stark et al., 2012*), Brazil (*Alonso et al., 2007*) and China (*Yu et al., 2013*)), but not in smaller countries (Israel (*Barnea et al., 2014*; *Huppert et al., 2012*)). Climatic covariates (*Alonso et al., 2007*; *He et al., 2013*; *Shaman et al., 2010*; *Yu et al., 2013*) and population size (*Bonabeau, Toubiana & Flahault, 1998*; *Stark et al., 2012*; *Viboud et al., 2006*) also appear to be important for epidemic onsets. A certain degree of consistency in the results obtained has been observed although studies have used a variety of methods and data: these are summarized in Table 1 (see Supplemental Information 1 for a discussion about the variability in data used).

From a methodological point of view, statistical methods applied for studying the impact of covariates on epidemic onset show important differences. Most studies have used a statistical approach (e.g., correlation tests (*Charaudeau, Pakdaman & Boëlle, 2014*; *Stark et al., 2012*) or regression models (*Crépey & Barthélemy, 2007*; *He et al., 2013*; *Yu et al., 2013*)). Only two studies (*Eggo, Cauchemez & Ferguson, 2010*; *Gog et al., 2014*) employed inference based on a dynamical model to study the factors affecting the geographical spread of the epidemic wave of two pandemics: *Eggo, Cauchemez & Ferguson (2010)* studied the 1918 Spanish Flu pandemic in England, Wales, and the US, and *Gog et al. (2014)* studied the 2009 H1N1 pandemic in England. A model was used in these studies that represented the rate (probability per unit of time) at which uninfected cities become infected according to covariates (such as the proximity of infected cities, city density or humidity).

Using a model inspired by classical dynamical models of infectious disease for statistical inference is appealing because such models attempt to capture the spread mechanism of pathogens. Such models have been employed for decades to represent the spread of

**Table 1 Summary of studies about influenza timing differences.**

| Where/scale | Data | Metric | Method | Results | References |
|---|---|---|---|---|---|
| EUSA/states | 30 years, weekly influenza-related mortality | Epidemic peak | Correlation tests | Correlation influenza spread/human movements (workflows) + influenza spread/population sizes | Viboud et al. (2006) |
| Pennsylvania, US/counties | 6 years, weekly laboratory confirmed influenza cases | Epidemic peak | Correlation tests | Correlation influenza spread/human movements | Stark et al. (2012) |
| France/departments | 25 years, weekly influenza syndromic cases | Epidemic peak | Correlation tests | Correlation influenza spread/human movements (school- and work-based communing) | Charaudeau, Pakdaman & Boëlle (2014) |
| France/patches 20 km | 8 years, weekly influenza syndromic cases | Epidemic peak | Correlation tests | Correlation number of influenza cases/density | Bonabeau, Toubiana & Flahault (1998) |
| Israel/cities | 11 years, weekly influenza syndromic cases | Epidemic peak | Statistical test | Highly synchronized epidemics | Barnea et al. (2014) and Huppert et al. (2012) |
| Brazil/states | 22 years, monthly influenza related mortality | Epidemic peak | Linear models | Spatial correlation suggesting a role of climate (temperature and humidity) | Alonso et al. (2007) |
| USA/states | 30 years, weekly influenza-related mortality | Epidemic peak | Correlation tests + linear models | Correlation influenza spread/air-traffic | Crépey & Barthélemy (2007) |
| France/regions | 20 years, daily influenza syndromic cases | Epidemic peak | Correlation tests + linear models | Correlation influenza spread/train- and automobile-traffic | Crépey & Barthélemy (2007) |
| China/provinces | 6 years, weekly laboratory confirmed influenza cases | Epidemic peak | Linear models | Strong correlation influenza spread/climatic factors (temperature, sunshine, rainfall), weaker correlation influenza spread/human movements | Yu et al. (2013) |
| Canada/provinces | 11 years, weekly laboratory confirmed influenza cases | Epidemic 25% quantile time | Generalized linear model | Correlation influenza spread/temperature, absolute humidity, population size and spatial ordering | He et al. (2013) |
| USA/states | 30 years, weekly influenza-related mortality | Epidemic onset | Correlation test | Correlation epidemic onsets/absolute humidity | Shaman et al. (2010) |
| USA/271 cities | 2009 H1N1 influenza pandemic weekly syndromic influenza cases | Epidemic onset | Correlation tests + mechanistic models | Strong correlation influenza onsets/school opening + short spatial diffusion, weaker correlation influenza onset/population sizes, absolute humidity | Gog et al. (2014) |

infectious agents (most often between individual hosts, but also between host populations (*Eggo, Cauchemez & Ferguson, 2010*; *Gog et al., 2014*; *Keeling, 2002*)). The second advantage is that, because the probability of entering into the epidemic state varies from week to week, epidemic onset dates can be linked to weekly variations of covariates. The use of dynamical modeling hence allows a deeper analysis of epidemic onsets than purely statistical models that try to establish a correlation between epidemic onset dates and the average value of covariates across the winter period (*Shaman et al., 2010*; *Yu et al., 2013*).

In the present paper, we have analyzed the impact of five covariates that could have potentially affected the time difference in the onset of epidemics between 18 regions of France over six epidemic years from 2006 to 2013 (an epidemic year corresponds to the period of time from October until the following April). The five covariates analyzed were temperature and absolute humidity, mobility flows, population size, and proportion of children within the region. Our study is based on a dataset provided by GROG (Groupes Régionaux d'Observation de la Grippe) an influenza surveillance network in France. The advantage of this network is that it combines clinical case definitions with identification of the virus. This is an important validation process because influenza can be clinically confounded with other co-circulating respiratory viruses.

Our analysis has the same modeling basis as *Eggo, Cauchemez & Ferguson (2010)* and *Gog et al. (2014)*. We put particular emphasis on the idea that the impact of a factor can be viewed at different scales that should be disentangled. For the studied covariates, we used permutation tests that overcome the problem of non-adjustment of the dynamic epidemic models (because not all factors that affect epidemic onset variability can be modeled). Indeed, by shuffling the observed values of covariates, we generate random (permuted) covariates that have no biological relation to the response variable (because they are random). Basically, if the observed value of a covariate performs significantly better than its permuted counterparts, this means that it is correlated to the response variable (even if the underlying model used in the analysis is not fully adjusted to the data).

## METHODS

### Data

In this analysis, the considered spatial scale is the region. The main reason for this is that the GROG network, from which the data originates, provides influenza prevalence estimates at the regional scale—so it was not possible to consider a lower scale here.

### *Epidemiological data*

Epidemiological data comes from the GROG network, a French surveillance network made up of voluntary general practitioners (GPs) and pediatricians. Sentinels record acute respiratory infections (ARIs) weekly and randomly send nasal samples for antigenic confirmation (or rejection) of influenza infection (see Supplemental Information 2 for more detail). Influenza incidence of clinical cases is then estimated as:

$$I_{\text{influenza}}(t) = I_{\text{ARI}}(t) \times T_+(t)$$

where $I_{\text{ARI}}(t)$ is the incidence of ARI cases and $T_+$ is the proportion of influenza-positive samples among ARI individuals. Details about the calculation of $I_{\text{ARI}}(t)$ and $T_+$ are given in Supplemental Information 2.

Epidemiological data are available from the epidemic years of 2006–2013 for all regions of metropolitan France (Fig. S1) except Languedoc-Roussillon, Franche-Comté and Limousin, where data were too scarce. Since we focus on seasonal epidemics, the 2009–2010 pandemic year was excluded.

For each year and region, we followed the GROG network procedure to define the epidemic onset:

1. Several similar influenza viruses (AH1N1, AH3N2 and B are considered different), more than what could be expected from the sporadic circulation of the virus that is observed at the beginning of the surveillance period, are detected or isolated in different areas of the same region.
2. At least two indicators (ARI reported by GPs + one of the five indicators: ARI reported by pediatricians, sick leave prescribed by GPs, GPs or emergency activity and drug distribution) increase by more than 20% compared to the average of October (of the season considered), without explanation by another phenomenon (i.e., no other local epidemic or outbreak due to other known cause).
3. A week is considered to be within an epidemic only if the previous or following week satisfies conditions 1 and 2. The epidemic onset date is defined as the first week that (i) satisfies 1 and 2 and (ii) is followed by a week satisfying 1 and 2.

Surveillance forms were routinely used during influenza seasons, and oral consent was obtained from each ARI patient when swabs were taken, in accordance with national regulations. All swab results and forms were anonymized by the laboratories before they were sent to the GROG network coordination, and only identified by a number given by each laboratory for virological tests. In accordance with the French applicable law, clearance by an Ethics Committee is not required in France for the retrospective analysis of anonymized data collected within routine influenza surveillance schemes.

### Mobility data

Flows of people generate contacts (including infectious ones) between populations from different regions. They can therefore promote influenza spread between connected regions and represent an important risk factor for regional epidemic onsets.

The National Institute of Statistics and Economic Studies (INSEE) provided mobility data in France. Place of residence and workplace are reported for employed individuals, while residence and school location are reported for students. We defined mobility flows as being journeys between home and work or school (Fig. 1). Note that these data are not representative of all possible journeys (e.g., vacations, weekends). Flows were only measured between regions and not at the lower scale (so, for example, travels from city 1 of region A to city 2 of region B and travels from city 3 of region A to city 4 of region B are considered to be equivalent in our analysis).

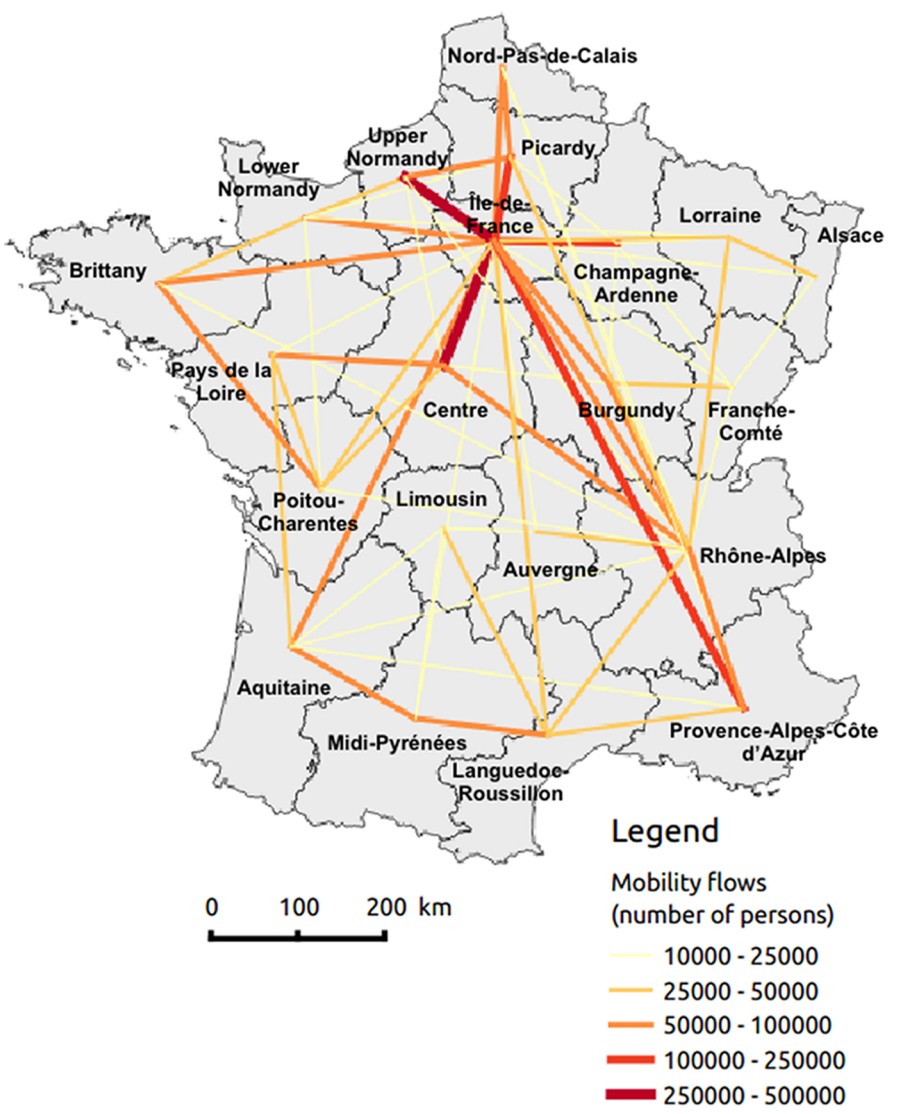

**Figure 1 Mobility flows by region made up with home–work and home–school journeys.**

### Demographic data

Favorable demographic characteristics of regions can also influence the spread of influenza and, hence, epidemic onset. We considered two demographic metrics (evaluated using INSEE data). The first metric is (the logarithm of the) population size, i.e., the number of individuals living in a given region, because contacts between individuals can be stronger in more populated regions, increasing the spread of the virus. We preferred considering population size instead of population density, as populations are not homogeneously distributed within regions (population density can be low due to large unpopulated areas despite cities aggregating many individuals). The second metric is the proportion of children from 0 to 19 years old, this age-class being the most affected by influenza and often suspected to be a major source of influenza transmission (*Wallinga, Teunis & Kretzschmar, 2006*; *White, Archer & Pagano, 2014*).

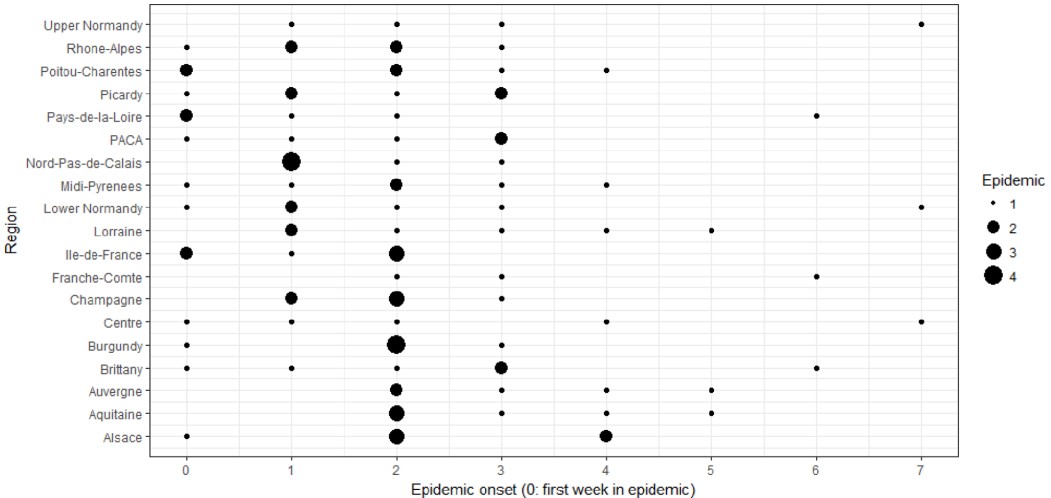

**Figure 2** Variations of epidemic onset dates (scaled each year so that 0 corresponds to the first week during which at least one region was in the epidemic state) between the 18 studied French regions. For all regions, we have six points (studied epidemic years), but note that some of these points might be overlapping.

### Climatic data

Climatic data were provided by Météo-France (the French national meteorological service). We selected 125 meteorological stations (Fig. S2) to estimate climatic covariates that globally describe the climate of each region. We focused on temperature and absolute humidity as climatic covariates. Even if they are correlated, they are both relevant as they might impact influenza epidemics (*Barreca & Shimshack, 2012*; *Roussel et al., 2016*; *van Noort et al., 2012*). Daily measures were averaged over the week and over the stations of a region to provide weekly variable metrics in all regions.

### Variability of data and covariates

Onsets of epidemics show variability at different scales (Figs. 2 and 3). At the **global scale**, epidemic onsets are more likely to occur during some weeks than others, whatever region or epidemic year is considered. At the **annual scale**, the average starting date (over regions) of epidemics varies between years. At the **spatial scale**, epidemics can start on average (over years) earlier in some regions than in others. Without additional sources of variability, we should expect to observe that some regions enter into an epidemic earlier in some regions every year and earlier during some years in every region than during others. In fact this is not the case, because local (a given year in a given region) specific winter conditions may change the timing of epidemics. This latter scale is termed **spatiotemporal**, because statistically it refers to an interactive effect of time and space on epidemic onset dates.

   To determine the scales at which epidemic onset dates and the different covariates exhibit a relevant amount of variability, we performed a preliminary analysis. Let us first consider the epidemic onset date variable. We used a linear mixed model with epidemic year and region as random effects. The distribution of the random effects are considered to be Gaussian, standard deviations being denoted $\sigma_Y$ and $\sigma_R$, respectively.

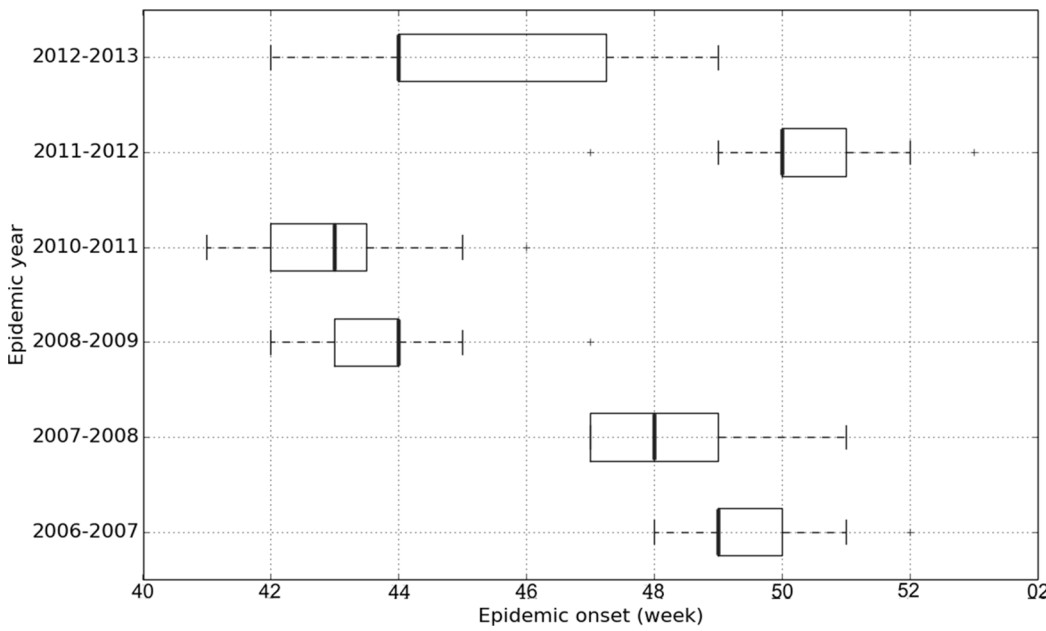

**Figure 3 Epidemic onset dates of French regions according to epidemic years given by the GROG network from 2006–2007 to 2012–2013 (except 2009–2010).** The 18 French regions serve as replicates for the boxplots of each epidemic year. 

This linear mixed model was performed with the R software (*R Core Team, 2016*) using the "lme4" package, using the following command line:

lmer(EpidOnset ∼ (1|Region) + (1|Year), data = FluOnsetData)

where FluOnsetData is the analyzed data set. Here the epidemic onset date was taken as a response variable (variable EpidOnset of the data set). Region and Year are the variables of the data set providing, for each observed epidemic, the associated Region and Year indexes (considered as qualitative variables), respectively.

A similar analysis was performed using demographic variables as variable responses, using the following command lines:

lmer(PopSize ∼ (1|Region) + (1|Year), data = FluOnsetData)
lmer(PropChild ∼ (1|Region) + (1|Year), data = FluOnsetData)

where PopSize and PropChild stand for the population size and proportion of children variables, respectively.

For climatic covariates, weekly data are available, so we added the week variable as a random effect in the linear model (the distribution of this random effect being also considered to be Gaussian, with a standard deviation denoted $\widehat{\sigma_{\mathrm{W}}}$), using the following line commands:

lmer(Temp ∼ (1|Region) + (1|Year) + (1|Week), data = FluOnsetData)
lmer(Humid ∼ (1|Region) + (1|Year) + (1|Week), data = FluOnsetData)

where Temp and Humid are the temperature and humidity variables in the data set and Week is the week index associated to each measure of these two climatic variables.
**Table 2 Preliminary analysis: evaluating the relevant scales of variation of the different variables (considered each separately) using the (preliminary) linear mixed model.**

| Factors | Intercept (average) | Regions (standard deviation, $\widehat{\sigma_R}$) | Years (standard deviation, $\widehat{\sigma_Y}$) | Weeks (standard deviation, $\widehat{\sigma_W}$) | Residuals (standard deviation, $\hat{\sigma}$) |
|---|---|---|---|---|---|
| Epidemic onset (week) | 6.95 | 1.50 | 1.69 | – | 3.83 |
| Population size (inhabitant) | 3,100,600 | 2,481,281 | 34,209 | – | 41,887 |
| Proportion of children | 0.24 | 0.014 | 0.002 | – | 0.001 |
| Temperature (°C) | 6.70 | 0.86 | 1.18 | 2.78 | 2.69 |
| Absolute humidity (g/m³) | 6.43 | 0.37 | 0.54 | 1.12 | 1.08 |

**Note:**
The importance of variations at the different scales is quantified by the corresponding estimated standard deviations (residuals and from random—regions, years and weeks—effects).

In total, five linear mixed models were performed (see command lines above). Regarding model outcomes, we used the "summary" function, which provides estimations for the residual variance (denoted $\hat{\sigma}$) and of the variance of random effects ($\widehat{\sigma_Y}$, $\widehat{\sigma_R}$ and $\widehat{\sigma_W}$ for climatic variables) for each of the five models performed.

For each of the five response variables considered, estimates of $\sigma_Y$ and $\sigma_R$ (and of $\sigma_W$ for climatic variables) provide a good descriptive tool to account for the magnitude of associated systematic variations at the different levels (systematic regional variations: $\widehat{\sigma_R}$, systematic inter-annual variations: $\widehat{\sigma_Y}$ and, for climatic variables, systematic variations between week: $\widehat{\sigma_W}$). Since we do not have replicates, for each of the five linear mixed models, residual variations of the model are confounded with the interaction between years and regions. For these reasons, $\hat{\sigma}$ quantifies the spatiotemporal standard deviation (i.e., how a given region/year deviates from what could be expected from the systematic effect of regions and years) of the associated variable.

The results of this preliminary analysis are summarized in Table 2. As epidemic onset dates vary at all scales, we can potentially relate their variation to covariates at all scales. Similarly, climatic covariates show important variation at all scales. Thus climatic covariates can be potentially linked to epidemic onset dates at all scales.

Demographic covariates can vary between regions but, in our data set, change very little between years. Hence trying to explain annual or spatiotemporal variation in epidemic onset with demographic covariates would be pointless in our case.

Mobility flows are not presented in Table 2. In practice, they are assumed to be constant in time. However, because we are interested in the mobility flows leading to virus exchange between regions, which depend on local influenza prevalences, the associated variable will vary at all scales and can be used to explain spatiotemporal variation in epidemic onsets. Therefore, we will try to determine whether flows leading to virus exchanges explain regional timing of an epidemic.

It is important to note that this preliminary analysis is completely independent of the main analysis that will be presented in the next section. The use of random terms (region, year and potentially, week) was important in this preliminary analysis because the objective was to quantify the variability of each variable at each scale. In the main
analysis, random terms will not be used because (i) they were not mandatory and (ii) they would render the model inference much more complex.

## Statistical methods

To analyze the link between epidemic onset dates and covariates, we used an approach based on statistical inference on a dynamical stochastic epidemic model. Due to the relatively small size of our data set, we reduced the number of parameters of the models as much as possible and avoided random (week, epidemic year or region) factors.

### The dynamical model

The dynamical model is a stochastic version of the Levin model adapted to the spread of infectious diseases within a metapopulation (*Keeling, 2002*) defined by the fact that, during a small time interval $[t, t + dt]$, the probability (for a non-infected region) of entering into the epidemic state for region $R$ during week $W$ of (the epidemic) year $Y$ is $\lambda(R, Y, W)dt$, where $\lambda(R, Y, W)$ is the rate at which a region enters into the epidemic state (the epidemic onset rate).

The epidemic onset rate is modeled as the product of two terms:

$$\lambda(R,Y,W) = \beta(R,Y,W) \times \phi(R,Y,W)^{\alpha}$$

where $\phi(R, Y, W)$ is (any quantity that is proportional to) the flow of virus entry within region $R$ during week $W$ of year $Y$ and $\beta$ is a proportionality term that can depend on $R$, $Y$ and $W$. The exponent $\alpha$ stands for the fact that the flow of virus entry might not affect the rate of epidemic onset in a linear fashion. For example, epidemic triggering could require the simultaneous presence of a sufficient number of infected individuals. In that case we would expect $\alpha$ to be greater than one because $x$ infected individuals during $n$ subsequent weeks are less likely to trigger an epidemic than $nx$ infected individuals during the same week.

### Mobility flows

Flows of virus entry are, to a large extent, related to flows of people between regions (i.e., mobility flows). Migration of the virus from region A to region B can be related to flows of people in both directions: individuals living in region A that contaminate individuals from region B during their travels and/or individuals from region B that acquire the infection during their travels in region A. To keep things simple, it is reasonable to assume that the probability that flows from region A will lead to an epidemic in region B with a rate that depends on (i) the number of people flowing between A and B and (ii) the proportion of people from A that are carrying the virus. Because symptomatic influenza alters the behavior of infected individuals (in particular their movement pattern), virus exchanges between regions are probably mostly ensured by asymptomatic individuals, but it is reasonable to assume that the number of asymptomatic individuals is proportional to the number of symptomatic (estimated by the GROG network).

As a result, the function $\phi$ is modeled as follows:

$$\phi(R, Y, W) = \sum_{i=1, i \neq R}^{N} (\delta_{Ri} + \delta_{iR}) \times \frac{I_i(W)}{S_i} + c \sum_{i=1, i \neq R}^{N} \frac{I_i(W)}{S_i}$$

where $\delta_{Ri}$ and $\delta_{iR}$ correspond, respectively, to mobility flows from region $R$ to region $i$ and from region $i$ to region $R$ (in number of people). $S_i$ represents the population size of region $i$ and $I_i(W)$ its incidence at week $W$ (thus $I/S$ is an estimate of the proportion of infected people). The term $\sum_{i=1, i \neq R}^{N} \frac{I_i(W)}{S_i}$ is the sum of influenza prevalence over all regions except $R$. We added this term because capturing the actual rate of virus exchange between two regions is complicated: the first term may be inaccurate and additional virus exchanges may originate from flows other than those modeled in this term. However, because we have no way of knowing where these exchanges come from, we did not make any distinction between regions (other than $R$) in this term. This is a classical assumption in epidemic metapopulation models, the first term corresponding to local transmission and the second to global transmission. $c$ is a positive constant parameter that quantifies the relative weight of local and global transmission. If the mobility flows we measured accurately capture the rates of virus exchanges between regions of France, then $c$ should be small.

### Climatic covariates

Let us consider a climatic covariate $X$ (temperature or absolute humidity) that takes the value $X_{R,Y,W}$ in region $R$, in year $Y$ and week $W$. To disentangle the four scales, we decompose $X$ into the sum of its mean value ($X_{\text{mean}}$) and four sub-covariates: $XW$, $XR$, $XY$ and $Xres$:

$$X_{R,Y,W} = X_{\text{mean}} + XW_W + XR_R + XY_{Y,W} + Xres_{R,Y,W}$$

where the $X$ will be replaced by any of the two climatic covariates ($X = T$ for temperature and $X = H$ for humidity).

The mathematical definition of the four sub-covariates and their biological interpretation are the following (please note that for all weekly averages, the average is calculated over the period starting in October of one year and ending in March of the following year).

$XW_W$ denotes the average value of $X_{R,Y,W} - X_{\text{mean}}$ over the different regions and the different years. $XW$ represents the overall (over all regions and years) global variation value of $X$. For example, if $TW = 4$, this means that the average temperature during week $W$ is 4 °C above the average value of the temperature over the epidemic period. Week $W$ is globally 4° warmer than the average. Because $XW$ measures the variations in the average temperature over weeks, it may explain variations in epidemic onset dates at the global scale (i.e., why epidemic onsets are more likely to occur some weeks than others). The objective here is to evaluate whether the average timing of influenza in the epidemic year is linked to average climatic conditions.

$XR_R$ denotes the average value of $X_{R,Y,W} - X_{mean}$ over the different weeks of the epidemic period and all years. $XR$ represents regional systematic differences. For example,

$TR = 2$ means that the average (over all weeks and years) temperature in region $R$ is $2°$ above the average temperature over all weeks, years and regions. Region $R$ is globally $2°$ warmer than the average. The sub-covariate $XR$ can explain epidemic onset variation at the spatial scale. The objective is to evaluate whether the time differences of influenza epidemic onsets between regions can be explained by different average climatic conditions between the regions.

$XY_{Y,W}$ denotes the average value of $X_{R,Y,W} - (X_{\mathrm{mean}} + XW_W)$ over the different regions. $XY$ stands for annual global differences. For example, $XY_{Y,W} = -5$ means that during year $Y$, the average temperature values that have been observed during week $W$ over all regions is $5°$ below the average values of temperature that have been observed over all regions and years during the same week $W$. If during year $Y$ all values of $XY$ are positive (during all weeks), this means that the winter of epidemic year $Y$ is globally warmer than the average. If $XY$ is negative during several subsequent weeks, it may reveal a cold snap in that period. Thus, $XY$ not only summarizes the average value of the covariate during the winter but also whether there have been some periods in the winter when the covariate was high and/or low (early epidemic onsets may simply arise from specific climatic conditions within limited time windows). It can explain variations of epidemic onset dates at the annual scale (i.e., why epidemics start on average earlier some years than others).

Finally, $Xres_{R,Y,W} = X_{R,Y,W} - (X_{\mathrm{mean}} + XW_W + XR_R + XY_{Y,W})$ represents spatiotemporal weekly residual variations. For example, $Tres_{R,Y,W} = -3$ means that, considering the average temperature values that where observed during week $W$ of year $Y$ in all regions on one hand, and the global characteristic of region $R$ compared to other regions on the other, the observed value of temperature in region $R$, week $W$ and year $Y$ is $3°$ below what could have been expected. So $Xres$ informs us about the local characteristics of a particular winter in each region and can be linked to variations in epidemic onset dates at the spatiotemporal scale.

### The complete model for $\beta$

The proportionality term $\beta$ can be different between regions, years and weeks because, considering a given flow of virus entry, local conditions within the region can, during a particular week, increase or decrease the risk of entering into an epidemic state. So $\beta$ can depend on several covariates, including demographic and climatic. The complete model (that integrates all the measured covariates) is defined by:

$$
\begin{aligned}
\log(\beta(R, Y, W)) = \ & a_0 + a_S \times \log(S_R) + a_C \times C_R + a_{TW} \times TW_W + a_{TR} \times TR_R \\
& + a_{TY} \times TY_{Y,W} + a_{Tres} \times Tres_{R,Y,W} + a_{HW} \times HW_W + a_{HR} \times HR_R \\
& + a_{HY} \times HY_{Y,W} + a_{Hres} \times Hres_{R,Y,W}
\end{aligned}
$$

where $S$ and $C$ represent respectively, the region population size and proportion of children. Note that since demographic covariates show little inter-annual variation, they are only likely to explain spatial variability in epidemic onsets. For that reason, we considered the average value of these covariates overall years in each region as model covariates. Parameters $a$ are model constant coefficients that quantify the link between each covariate and $\beta$. To allow a direct comparison between all the coefficients $a$, the four

covariates ($S$, $C$, $T$ and $H$) have been centered and standardized before the analysis. Coefficient $a_0$ is the intercept of the model.

## Model likelihood

Model parameters were estimated using a maximum likelihood procedure. The link between epidemic onset dates and model covariates was tested using the likelihood-ratio test (LRT) statistic. The chi-square approximation of the LRT was not used here because it requires both large sample size and assumes that data can be considered as a plausible outcome of the model (i.e., model adjustment). In our case, model adjustment requires all potential sources of weekly, inter-annual and inter-regional variations to be incorporated in the model. Because this was not the case—we did not include random terms in our model—we preferred not to rely on this approximation. Instead, permutation tests were used (see below).

For an epidemic year $Y$, the probability of a region $R$ to enter into an epidemic state in a particular week $W$ is given by the probability that the region did not enter into an epidemic state before week $W - 1$ : $e^{-\sum_{i=0}^{W-1} \lambda(R,Y,i)}$ and the probability that the epidemic occurs during the week that started at $W$ : $1 - e^{-\sum_{i=0}^{W-1} \lambda(R,Y,i)}$. That is why the likelihood ($L$) of a region $R$ and an epidemic year $Y$ is defined as:

$$L = e^{-\sum_{i=0}^{W-1} \lambda(R,Y,i)} \cdot \left(1 - e^{-\lambda(R,Y,W)}\right)$$

The global likelihood ($L_g$) is defined as the product of the regional likelihoods for each epidemic year, given by:

$$L_g = \prod_{R,Y} e^{-\sum_{i=0}^{W-1} \lambda(R,Y,i)} \cdot \left(1 - e^{-\lambda(R,Y,W)}\right)$$

Model parameters were inferred using maximum likelihood estimation. Models and permutation tests were implemented in Matlab.

It should be noted that, due to an insufficient covering during some weeks in some regions, influenza incidence could not be estimated for these points. Because the statistical procedure requires incidence values to calculate the terms associated with mobility flows, we replaced missing incidence values by zeros in the program.

Among the 107 observed regions/years, five did not show any epidemic. Including these data points in the analysis is feasible (under its current form, the Matlab code integrates this possibility). However, including them altered the results of the analysis in a way that we think is counterproductive (see Supplemental Information 3 for more details), so we preferred to exclude them from the analysis. From a biological point of view, this choice is reasonable because it is likely that these regions/years present specific characteristics (e.g., an important proportion of immune individuals) meaning that, despite an important flow of virus entry, they could not enter into the epidemic state. This case scenario was not integrated in the model, which assumes that, provided a sufficient flow of virus entry, any region could enter into the epidemic state during any season.

**Table 3 Summary of the studied covariates (whose link with epidemic onset dates was tested) with associated sub-covariates, model parameters, scales of variation and indexes permuted.**

| Covariate | Sub-covariate | Associated parameter | Scale | Permuted index |
|---|---|---|---|---|
| Temperature | $TW_W$ | $a_{TW}$ | Global | Weeks |
| | $TR_R$ | $a_{TR}$ | Spatial | Regions |
| | $TY_{Y,W}$ | $a_{TY}$ | Annual | Years |
| | $Tres_{R,Y,W}$ | $a_{Tres}$ | Spatiotemporal | Regions and years |
| Absolute humidity | $HW_W$ | $a_{HW}$ | Global | Weeks |
| | $HR_R$ | $a_{HR}$ | Spatial | Regions |
| | $HY_{Y,W}$ | $a_{HY}$ | Annual | Years |
| | $Hres_{R,Y,W}$ | $a_{Hres}$ | Spatiotemporal | Regions and years |
| Mobility | $\sum\limits_{i=1,i\neq r}^{N}(\delta_{ri}+\delta_{ir})\times\frac{I_i(t)}{S_i}$ | – | Spatiotemporal | Regions |
| Population size | $S_R$ | $a_S$ | Spatial | Regions |
| Proportion of children | $C_R$ | $a_C$ | Spatial | Regions |

## Permutation tests

Permutation tests are based on the idea that randomly shuffling the values of a covariate $F$ looks at the distribution of the possible linkages that could have been found between $\lambda$ and the covariate $F$ given data. Hence, replicates of random shuffling of the values of $F$ can be used to estimate the distribution of the LRT under $H_0$ "no impact of the covariate" (*Lebreton, Choquet & Gimenez, 2012*). An interesting property of covariate (rather than data) shuffling is that other covariates can remain unshuffled and keep their ability to reduce residual variance.

Because several covariates vary according to only one index ($W$, $R$ or $Y$), we used block permutations—covariates were shuffled according to some indexes but not others—to keep the error structure of covariates. For example, population size ($S$) varies only between regions. Hence, the associated permutation test shuffles the values of $S$ between regions but keeps it constant between weeks and years. According to their scale of variation, all covariates were tested according to a specific set of indexes (Table 3).

The four following steps can summarize the principle of permutation tests:

Step 1: shuffle randomly a covariate. Potentially, variables have three indexes of variations: weeks ($W$), year ($Y$) and region ($R$). Let us call $P$ a random permutation of the triplet ($W$, $Y$, $R$) (the different types of permutation that can be used will be detailed below). Let us call $X$ the covariate that has to be permuted. The original (non-permuted) covariate is $X_{W,Y,R}$. The permuted covariate is called $Z$ and is defined by $Z_{W,Y,R} = X_{P(W,Y,R)}$.

Step 2: determine the test statistics associated with each permutation. We used the LRT, defined as $-2 \times \log\left(\frac{L_Z}{L_0}\right)$, where $L_Z$ and $L_0$ respectively represent the likelihoods of models with and without covariate $Z$. Note that, for mobility flows, the model without this term is not used (the associated coefficient always equals one). In that case, the

LRT statistic used is replaced by the deviance (defined by $-2\log(L_Z)$) statistic, other steps being unchanged.

Step 3: determine the distribution of the LRT statistic under the null hypothesis $H_0$: "epidemic onsets are independent of covariate $X$." Since permutations generate random covariates that have no biological reason to be associated with epidemic onsets, each permutation represents a random realization of the LRT statistic under $H_0$. For each covariate $X$, 1,000 permutations were generated and Steps 1 and 2 led to 1,000 independent values of the LRT under $H_0$. From that we could derive an estimate of the distribution of the LRT under $H_0$.

Step 4: determine a threshold for the LRT under $H_0$. The threshold was simply taken as the 95% quantile of the distribution of permuted LRTs. Comparing the observed value of the LRT with this threshold provides a test criterion for rejecting, or not, $H_0$.

Alternatively, we can estimate a $p$ value for each test, defined as $p = (x + 1)/(N + 1)$, where $x$ is the number of permuted values of the LRT above that observed and $N = 1,000$ is the number of permutations. $H_0$ is then rejected as soon as $p < 0.05$ but is otherwise accepted.

Based on the level at which we want to establish correlates between epidemic onset dates and covariates, different tests have to be performed. If we want to test a covariate that explains epidemic onset variations at the spatial level, only region indexes will be shuffled. In practice, let us call $P_R$ a permutation of region indexes, then a permutation shuffling only regions indexes will take the form of $P(W, Y, R) = (W, Y, P_R(R))$. Shuffling only region indexes means that measures are repeatedly the same each year and each week within a region.

Similarly, shuffling only year indexes will test covariates explaining annual variations in epidemic onsets. Let us call $P_Y$ a permutation of years, the permutation taking the form: $P(W, Y, R) = (W, P_Y(Y), R)$. In the same way, shuffling week indexes will test covariates explaining global variations (why epidemic onset does not happen randomly within the studied period). By calling $P_w$ a permutation of the week, the permutation will take the form $P(W, Y, R) = (P_W(W), Y, R)$.

For climatic covariates explaining spatiotemporal variations in epidemic onsets, we chose to independently shuffle region and year indexes. In practice, the permutation will take the form of $P(W, Y, R) = (W, P_Y(Y), P_R(R))$. Shuffling region and year indexes independently rather than simultaneously has the advantage of keeping the general intra-annual and intra-regional structures in covariates.

Finally, for the mobility covariate permutations, we first shuffled regions (in the $\delta$ matrix, similar permutations were used for lines and columns of the matrix) and then recalculated the (permuted) flow of people between all pairs of regions (coefficients $\delta$). Then the flow of infected people was calculated by multiplying these coefficients by the non-permuted regional prevalence, leading (for all regions, years and weeks) to a new value for the first term of $\phi$ (i.e., $\sum_{i=1, i \neq R}^{N} (\delta_{Ri} + \delta_{iR}) \times \frac{I_i(W)}{S_i}$). The advantage of this choice is that it tells us how re-associating regions randomly explains the observed synchrony between connected regions. Permuting the region indexes allows us to keep

the structure of the global connection network of the country (e.g., the fact that some regions are more connected to other regions than others). In summary, the connection network between the regions remains the same in permuted data but their link to epidemic onset probabilities is broken.

One important question when testing the link between a response variable and covariates is the set of correction covariates that should be introduced. One way to deal with this question is to use the complete model and remove the covariate we want to test. This solution is interesting because, if the test turns out to be significant, then the link between the response variable and the covariate that is observed cannot be explained by any confounding effect of the other covariates. Considering our relatively low sample size, this is not the solution we retained here because it is conservative, especially when covariates are correlated (which is, e.g., the case for temperature and humidity). Instead, for each covariate, the link was tested without correcting by all the covariates that have the same scale of variation. The other covariates were kept because they can capture some of the epidemic onset date variability.

The case of mobility flow is singular because this variable is included as a correction covariate in all models and it is not associated with any model parameter. Permutation tests were also performed on this covariate (see above). We performed two different tests. In the first (termed "corrected") we kept all other covariates as correction terms (so we use the complete model). In the second (termed "uncorrected"), we removed all the other (demographic and climatic) covariates.

## RESULTS

The main model parameters (that quantify the impact of the studied covariates) are given in Table 4, together with the associated $p$ value of the corresponding test. A table summarizing all the model parameters inferred from all the different models used can be found in Table S1. Covariates are considered to be significantly linked to epidemic onset dates as soon as the associated $p$ value falls below 5%. Figures showing the distribution of the LRT statistic are given in Figs. S3–S6.

Absolute humidity was found to be significantly linked to epidemic onset dates at the spatial scale ($p = 0.029$), but not at the other scales. The associated coefficient was negative ($-0.4763$).

Mobility flows were not found to be significantly linked to epidemic onset dates ($p = 0.57$ with the corrected model, $p = 0.73$ with the uncorrected model). In the corrected model, the coefficient associated with global incidence was very high, even when we considered that the local transmission term was multiplied by mobility flows (whose average is around 14,400). Such an important weight of the global incidence is not found in the uncorrected model were we removed all covariates (although the test of mobility flows remained not significant, see Table S1). This suggests that the combination of covariates used in the complete model best explains spatiotemporal variation than those explained by mobility flows.

Population size and proportion of children were not significantly linked to epidemic onset dates at the spatial scale.

**Table 4 Estimates of the associated parameter tested for each covariate with the *p* value of the associated permutation test.**

| Covariate | Symbol | Estimate | *p* Value |
|---|---|---|---|
| T: global | $TW_W$ | −0.4932 | 0.1718 |
| T: spatial | $TR_R$ | −0.2557 | 0.1598 |
| T: annual | $TY_{Y,W}$ | −0.3841 | 0.2627 |
| T: spatiotemporal | $Tres_{R,Y,W}$ | 0.0461 | 0.9361 |
| H: global | $HW_W$ | −0.0200 | 0.1089 |
| H: spatial | $HR_R$ | −0.4763 | 0.0290 |
| H: annual | $HY_{Y,W}$ | −0.0449 | 0.7512 |
| H: spatiotemporal | $Hres_{R,Y,W}$ | −0.3004 | 0.7932 |
| Mobility flows: corrected | $\sum_{i=1,i\neq r}^{N} (\delta_{ri} + \delta_{ir}) \times \frac{I_i(t)}{S_i}$ | – | 0.5704 |
| Mobility flows: uncorrected | $\sum_{i=1,i\neq r}^{N} (\delta_{ri} + \delta_{ir}) \times \frac{I_i(t)}{S_i}$ | – | 0.7333 |
| Population size | $\log(S_R)$ | 0.1274 | 0.1718 |
| Proportion of children | $C_R$ | 0.1215 | 0.0929 |

Note:
For each covariate, all these pieces of information come from the model used to evaluate the link between the covariate and epidemic onset dates.

# DISCUSSION

We have presented an approach inspired by the dynamical modeling presented in *Eggo, Cauchemez & Ferguson (2010)* and *Gog et al. (2014)* to test and quantify the link between several covariates and the onset date of epidemic influenza in France. The objective was both to provide new insights in influenza epidemic knowledge and, more generally, to discuss the issue of the multiple scales by which the link can be viewed and propose permutation tests associated with each level of variation.

## Impact of mobility flows and demographic covariates

Our results did not reveal an impact of mobility flows on epidemic onset dates. This is quite surprising because mobility flows of infected individuals between regions can help the accumulation of a critical number of infected people leading to the influenza outbreak. Previous studies showed a correlation between daily work commutes and global influenza spread as well as regional epidemic peaks in France (*Charaudeau, Pakdaman & Boëlle, 2014*; *Crépey & Barthélemy, 2007*) and also in USA (*Crépey & Barthélemy, 2007*; *Stark et al., 2012*; *Viboud et al., 2006*). The fact that we did not observe this link in our study may be due to inaccurate estimates of these flows. Simply considering flows of workers and students (and not those linked to holidays and week-ends) could be too simplistic. The spatial scale at which we worked (the region) could also be too narrow to view the spatial spread of the virus.

Children are also central to the spread of a disease like influenza. They are the most aggregated age-class of the human population and have a relatively naïve immune system (in terms of immune memory). Consistently, several studies (*Peters et al., 2014*; *Schanzer, Vachon & Pelletier, 2011*; *Stockmann et al., 2013*; *Timpka et al., 2012*) have

reported earlier epidemics in school-age children than in other age groups. Furthermore, in England (*Pebody et al., 2015*) and in Florida (*Tran et al., 2014*), vaccination of school age children has been shown to reduce influenza incidence in all age-classes as well reducing excess respiratory mortality, stressing the role of children in influenza transmission. We have not found any statistical association between demographic covariates and epidemic onset dates.

## Climatic covariates: a typical example of a multi-scale issue

Climate is also an important factor for virus spread. It affects virus survival outside the host (*Lofgren et al., 2007*; *Lowen et al., 2007*), host susceptibility to the infection (*Eccles, 2002*) and human behavior (*Lofgren et al., 2007*). Studying its impact on influenza epidemic onsets is hence relevant, but as it can be viewed at different scales, its analysis is more complex.

In eco-epidemiology (and in ecology in general), it is more and more common to deal with data acquired at multiple scales (spatial, temporal, populational, individual, etc.). Such data present a methodological challenge because covariates may explain the variability of data at different scales. In our example, epidemic onsets showed four levels of variability. At the highest level (global), climate may explain why influenza epidemics occur more frequently in some weeks than in others. At the spatial scale (respectively, annual), they may explain why influenza epidemics start earlier on average in some regions (respectively, years) than in others. At the lowest scale (spatiotemporal), local climatic conditions could explain why an epidemic occurs earlier or later in a given year in a given region.

In general, larger scales are associated with the more confounding effects. Systematic changes in climate between regions also come with systematic changes in other covariates (such as demography, economy, etc.). Similarly, systematic shifts in climate between years come with shifts in, e.g., antigenic characteristics of influenza strains, human society characteristics (that evolve in parallel with climate changes). All these covariates can introduce statistical confusion in the interpretation of model inference.

The smallest scale, where we try to link deviations in epidemic onset with deviations in climate (after accounting for systematic variations in yearly and regional average climate), would in our case be the ideal statistical scale. However, it also comes with more noise in variable estimates, which is reduced at the upper scales (which are averages).

The only scale at which the impact of climate was found to be significant here was the spatial scale for humidity. This means that, in region with dry climates, epidemics of influenza tend to start earlier. However, the *p* value associated with this covariate was close to 5% and one could wonder whether the link could be artificial considering the number of tests we performed in our analysis. In any case, it is interesting to note that, for all climatic covariates whose coefficient was not close to zero, all values were negative, which is consistent with the idea that dry and cold climates promote the spread of influenza.

## Methodological issues

Dynamical modeling offers a natural basis for understanding the spread of infectious diseases. Paired with statistical tools, they have been used with success to analyze the spread of infectious agents within non-spatialized (*Chowell et al., 2004*; *Gibson, Kleczkowski & Gilligan, 2004*) as well as spatialized (*Fang et al., 2016*; *Gibson, 1997*; *Merler et al., 2015*) host populations. However, because they are based on the modeling of the mechanisms underlying the spread of agents, such approaches raise important methodological issues.

Linking the probability of epidemic onset to weekly shifts in climatic covariates is appealing but requires accurate onset date estimates. Because the climate can change rapidly during the winter in France, a lag of a few weeks between the real and observed onset dates weakens the strength of its link with climatic covariates. The major difficulty with observational estimates of epidemic onset dates is that they are based on a clinical criterion (atypical increases in influenza infection). If this choice is legitimate from a management point of view, it does not necessarily translate the real epidemiologic point when all conditions are gathered to ensure the massive spread of the disease and a time lag may exist between this "break point" and the estimated point.

Another important point regarding epidemiological models is that, at least in our case, they cannot perfectly describe the variability of the response variable. This would require capturing all the variations of the probability of epidemic onset between weeks, years and regions. Within a simple dynamical model, it is unfortunately not possible to account for all the complexity of the transmission process. Vacations were not included in the analysis. Integrating them would have been complex because, in France, regional vacations are not synchronized. Vacations affect the spread of a virus like influenza in a complex way (*Cauchemez et al., 2008*). Schools are closed and travel patterns are changed, and travel associated with work or study is replaced by tourism. Unfortunately, we had no such fine information in our data set.

Network coverage was also an important issue. Three regions could not be studied for this reason and, in others, we had some points missing in our prevalence estimates. This can have implications for the estimate of virus entry within regions, missing points being potentially associated to unquantified flows of virus entry. However, because missing data were mainly associated with poorly connected regions and/or to periods of the year when influenza prevalence is low, we believe that neglecting them is not too prejudicial for the analysis.

It is important to remind that, for some epidemic years in some regions, no epidemic of influenza was observed. For reasons detailed in Supplemental Information 3, we chose to remove these regions from our analysis. This implies that our results are only relevant for understanding the link between influenza epidemic onset dates and covariates for regions and epidemic years for which an epidemic did occur and should not be extrapolated to explain why no epidemic occurred in some circumstances.

Another important point to discuss in such an analysis is the geographical scale at which data are measured. Due to the spatial covering of the GROG network, it was

not possible to work below the regional level. We are conscious that many phenomena may occur at lower scales: regions are not homogeneous in terms of human density, movement patterns and climate. However, because this problem is due to the basic structure of the data, there was not much we could do.

For all these reasons it was important not to rely on the asymptotic assumption of the chi-square distribution of the likelihood ratio statistic. Such an assumption is only valid when the model is able to describe the complexity of the variations of the response variable (here the epidemic onset rate). Here, this would have been a very strong assumption, as we can see on Figs. S3–S6 (where the 95% rejection thresholds are quite different from what we would have observed with a chi-square approximation of the likelihood ratio statistic). In such a context, permutation tests appear to be a very interesting tool to overcome the issue of model adjustment. Indeed, permutation tests of covariate focus on the distribution of the covariate (which is simple) and not on that of the response variable (which is complex). Thus, even if the underlying model is incorrect, permuted covariates have absolutely no reason to perform better than those observed. They offer therefore, a robust means to test the impact of the different covariates.

If permutation tests reduce the risk related to robustness of the analysis to depart from model assumptions, they also have some drawbacks. They require a lot of computation time to perform a large number of permutations, each one requiring involving the recomputation of the test statistic. Also, they consider fixed observed values for all the variables, evaluating whether the pattern observed in the data is likely, or not, to have arisen by chance. The underlying theory of permutation tests is hence not based on the random sampling assumption (made in parametric approaches), which has the advantage that the conclusions of the analysis can be generalized to the entire population (*Ernst, 2004*). So in contrast, from a theoretical point of view, permutation tests only allow one to draw conclusions that are relevant to the particular data set.

In addition, permutation tests do not resolve the important problem of statistical power. The data set we analyzed here is relatively small (around a hundred points). Because our approach is relatively new, it is hard to know whether such a data set is sufficient for a reasonable statistical power.

The lack of statistical power is probably the reason why we found so few associations in our analysis. Therefore, it is important to note that our inability to detect effects is far from proving their absence. We believe that our study suggests a novel means to treat epidemic onset data by combining dynamical modeling with hypothesis testing based on permutation tests of the covariates.

Testing the significance of the observed associations is already a complex task by itself, so in the present paper we chose not to address the issue of evaluating confidence intervals for our model parameters. In our case, such intervals would not be very insightful because we found only one significant association (with a *p* value that is close to the rejection threshold, raising the question of multiple testing effects).

As a future direction, permutation tests provide an interesting way to evaluate equivalents of confidence intervals (*LaMotte & Volaufova, 1999*). Such intervals are

quite complex to implement and are still marginal in the literature but present the advantages of permutation tests that we exposed earlier.

## Link with the survival analysis approach

Using dynamical modeling may appear rather complex to non-methodologists because of the lack of existing software packages to implement such models. Handmade programs are also exposed to programming mistakes. Although we carefully checked our program, such mistakes could not be excluded.

For people who (arguably) prefer methods based on long-term existing software packages, an interesting comparison can be made between our approach and (Cox regression) survival analysis models. The modeling basis of both approaches are the same. The rate of epidemic onset is similar to the hazard function. Cox regression uses linear links between the logarithm of the hazard function and covariates. Our link is slightly more complex, the only source of non-linearity lying in the fact that we sum the local and global flows of virus entry. Here, linearization of the relationship between the logarithm of the epidemic onset rate and covariate could be achieved with only a few approximations.

However, it is important to note there is an important difference between our analysis and Cox regression survival analysis that involves the way in which likelihood is calculated. Cox regression uses partial likelihood. Basically, partial likelihood consists of comparing the value of covariate every time an event occurs. Thus, the Cox regression model finds the best linear combination of covariates that maximize the probability that, considering that several events could have occurred on a given date, the observed event (associated with the date) was the one that occurred. So, partial likelihood does not try to explain why events occurred on the precise date that they did occur but why they occurred in a given order.

In contrast, the way we calculated likelihood here integrates this information. So for example, if an epidemic onset occurred at the beginning of December in a given region during a given year, our method tries to find the combination of covariates that best explains why the onset did not occur earlier (e.g., by trying to link it to specific climatic conditions that were present at the beginning of December but not in November). This is quite different from what is done with the partial likelihood of the Cox regression.

Which way of calculating likelihood is better is still unclear due to the absence (to our knowledge) of theoretical studies comparing both approaches. It is all a matter of which pieces of information we want to include to infer model parameters. The Cox regression has the advantage of being implemented in many classical software routines of data analysis (such as R). Thus, for researchers who are inspired by our approach to analyze epidemic onset data, adapting our model (basically by linearizing the relationship between the logarithm of the epidemic onset rate and covariates) to the Cox regression framework could represent an interesting compromise to overcome the programming issues associated with our approach.

# ACKNOWLEDGEMENTS

We acknowledge the practitioners of the sentinel network Réseau des GROG and the labs involved in the surveillance. We thank Isabelle Daviaud from Open Rome who sorted the epidemiological data from the sentinel network Réseau des GROG and Annick Auffray from Météo-France for her kind help with meteorological data. We thank Robin Buckland for proofing the English manuscript. This work was archived using the computing facilities of CC LBBE/PRABI and CC IN2P3. It was performed within the framework of the LABEX ECOFECT (ANR-11-LABX-0048) of the University of Lyon, within the program "Investissements d'Avenir" (ANR-11-IDEX-0007) operated by the French National Research Agency (ANR).

### Funding

The authors received no funding for this work.

### Competing Interests

The authors declare that they have no competing interests.

### Author Contributions

- Marion Roussel conceived and designed the experiments, performed the experiments, analyzed the data, contributed reagents/materials/analysis tools, prepared figures and/or tables, authored or reviewed drafts of the paper, approved the final draft.
- Dominique Pontier conceived and designed the experiments, performed the experiments, analyzed the data, contributed reagents/materials/analysis tools, authored or reviewed drafts of the paper, approved the final draft.
- Jean-Marie Cohen conceived and designed the experiments, contributed reagents/materials/analysis tools, authored or reviewed drafts of the paper, approved the final draft, specialist in influenza surveillance.
- Bruno Lina conceived and designed the experiments, contributed reagents/materials/analysis tools, authored or reviewed drafts of the paper, approved the final draft, virologist specialist in influenza virus.
- David Fouchet conceived and designed the experiments, performed the experiments, analyzed the data, contributed reagents/materials/analysis tools, authored or reviewed drafts of the paper, approved the final draft.

### Human Ethics

The following information was supplied relating to ethical approvals (i.e., approving body and any reference numbers):

Surveillance forms were routinely used in the influenza seasons, and oral informed consent was obtained from the ARI patient at the moment of swab taking in accordance with national regulations. All swab results and forms were anonymized by the laboratories before they were sent to the GROG network coordination, and only identified by the number given by each laboratory for virological tests. In accordance with the French

applicable law n°2011–2012 of the 29th December, article 5, no clearance of an Ethics Committee is required in France for the retrospective analysis of anonymized data collected within routine influenza surveillance schemes.

## Data Availability

The raw data and code have been supplied as Supplemental Dataset Files.

## Supplemental Information

Supplemental information for this article can be found online at http://dx.doi.org/10.7717/peerj.4440#supplemental-information.

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
