# Peer review of "Linking influenza epidemic onsets to covariates at different scales using a dynamical model"

_PeerJ, doi:10.7717/peerj.4440_

## Round 0.1 · original submission · Major Revisions

Your manuscript looks suitable for review but you have uploaded no source code, as required by the journal. Before we send it for review, please provide this code. Thank you.

---

## Round 0.2 · Major Revisions

Both of the referees had suggestions for revisions and I would like to see you address these if you choose to resubmit. As far as I can perceive these suggestions are not contradictory or mutually-exclusive and therefore the list of revisions for you is long.

I would like to highlight a few points, but note that your response document needs to address (in some fashion) each of the referees' points, not only these.

One of the referees notes that a more user-friendly data format should be supplied. I agree.

One of the referees notes that the paper might be cast in a more exploratory light, given the sample size. This on the face of it seems like a reasonable suggestion so I encourage you to consider it or to provide a convincing rebuttal.

One of the referees feels the manuscript would be improved by a thorough editing for miscellaneous English-language issues. I endorse this. The manuscript is comprehensible but could be improved in this area.

One of the referees is puzzled by the modeling of the factors in groups (vs all at once). So am I.

·

Basic reporting

- The paper is written in professional English throughout. There are a few minor grammatical errors (e.g. line 23 the use of “the same factor”, line 71 the use of “on” and line 83 the use of singular “agent” and a few others) but all very minor.
- The raw data is supplied but would be much better if it were in a standard machine-readable format like .csv or .txt. Having it in a PDF file is unusual and a bit annoying for anyone wishing to replicate the analyses. Ideally, the data could be in the .zip file containing the code with the same name as is used to import in the code. As it is, I don’t think the analysis is reproducible at least in part because the data cannot be easily read in.
- In Figure 1 (and perhaps Figure 2 as well, although I feel less strongly about it here) it would be better to show the raw data. The boxplot is summarizing six data points with five and giving a false sense of a large dataset and estimated std deviation when there is actually very little data per row.
- Overall the literature review is thorough.
- Subscripts on lines 204 and beyond are not consistent. Are years subscripted with S or Y?

Experimental design

- The rules for defining onset are not sufficiently precise or well-defined. E.g. What is several? More than 1? What does without explanation by another phenomenon mean? What if the week itself meets criteria 1 and 2? Under this definition, it would not be included because the previous and following week do not meet the conditions?
- The most severe limitation of this study is the limited number of onsets (19 regions and 6 years, so just over 100 total observations). Given this, the authors are estimating a large number of coefficients from this dataset, in Table 4 there are 11 coefficients, which means barely 10 observations per coefficient. And other coefficients are fit and interpreted as well.
- It is not statistically justifiable that this study has power to detect associations from as complex a model as this with as little data as they have. This is a larger concern given the complexity and multiple stages of the modeling that occur.
- The “holiday effect” has been demonstrated to have an impact on flu dynamics in the US, so it seems like a major limitation that it is not incorporated here (line 145). (see, e.g. http://dx.doi.org/10.1371/journal.pcbi.1004382)
- Is it a problem that the model (on line 222, e.g.) may be overly constrained by not having an intercept?
- Some data appears to be missing (seen visually in the graphs). How is this dealt with in the analysis?

Validity of the findings

- The authors spend quite a bit of time extolling the virtues of their approach but not enough time acknowledging the limitations. E.g. lines 81-89 discuss advantages of dynamic models without discussing their disadvantages (one that comes to mind is that they can be mis- or over-specified).
- There is little discussion of the major limitation of the study, which is the sample size. More needs to be said about this, relative to the strength of the conclusions that are being drawn. How can we be confident in our conclusions with such a heavily adjusted analysis with such few datapoints?
- I do not see how the conclusion on lines 367-369 are justified by the results shown. At best it is an over-confident statement.

Additional comments

Overall, my major concerns with this article revolve around the sample size and the complexity and quantity of the fitted models. I am concerned that many different inferences are being drawn from multiple multi-stage analyses of a small dataset, with no adjustment for multiple comparisons. Since the method is somewhat new and specialized, there is little intuition or formal work that we can rely on to understand how much power these models could be expected to have to detect true associations in the data.

Reviewer 2 ·

Basic reporting

One of my biggest criticisms of this paper is the writing. I realize that English is not the first language of the authors; I am sympathetic and can only imagine what an article I wrote in French would be like. That being said, I would strongly encourage the authors to seek out the help of someone who is a native English speaker or some sort of professional editing service. I am attaching a marked-up version of the manuscript with places where I noted problems with phrasing, word choice, etc, however, it is not an exhaustive list of corrections. The writing/language issue was severe enough that it was difficult to interpret what the authors were trying to communicate at times. One acute example of this is the word "variations" that the authors use frequently. I think in some cases, like when the authors discuss decomposing covariates, the appropriate word would be "deviations" (that would be the standard statistical word choice). In other cases, I think the authors actually mean "variability" when "variations" is used. The word "variations", in my opinion, indicates several different realizations of something (e.g. "variations on a theme"), and I don't think the authors ever intend the word in this manner in this manuscript. Anyway, this is just one example of many from the paper (again, see the marked up PDF).

The literature discussed by the authors is relevant but somewhat limited in scope. Analyses such as the one being presented here have been used for other disease, most recently the spread of Ebola in West Africa where researchers were interested in when Ebola might reach a particular region (e.g Merler et al. 2015, http://dx.doi.org/10.1016/S1473-3099(14)71074-6, or Fang et al. 2016, http://dx.doi.org/10.1073/pnas.1518587113). The authors should discuss their modeling strategy in the broader context of other spatiotemporal modeling strategies that are used for non-influenza diseases (Table 1 covers many influenza models).

Figure 1 is a bit hard to read. I would suggest compressing the x-axis to allow for bigger labels on the axes in order to increase legibility. The y-axis could be expanded. Also, because there are only six data points per regions plotted, it would be better to just show the data points and not box plots (which obscure the actual distribution of the data).

The decomposed variable labels (e.g. "OWV") should be shortened to a single letter or symbol. It would be okay to have something like OWV if it some type of acronym (the authors state it is the "overall global variations"), but even then it would be preferable to use a symbol with some kind of systematic sub/super-scripting.

Along similar lines, way too many of the modeling details are placed in the appendix. It was quite hard to assess the models because of a continuous need to go and look at the appendix. Given that there are no space limitations in PeerJ, the reader should be able to sufficiently assess the modeling strategy by simply reading the body of the manuscript and not need to read the appendix to understand what is being done.

It is mentioned in several places (e.g. line 32, line 81, etc) the authors suggest that they are using a "dynamic modeling" approach. I think this is a bit misleading. Dynamic modeling implies that ODEs/PDEs are being used in the modeling; there really is none of that in the model presented here. Just because disease incidence and movement patterns are being used to make statistical inference doesn't mean that the model is dynamic. As a reader, I was disappointed when there was not more of "dynamic" component to the modeling.

Experimental design

I have some concerns about the modeling that has been done. First and foremost, it is unclear to me why the authors chose not to build a model that included all covariates simultaneously. Rather, they have 4 different models (based separately on temperature, humidity, mobility, and demography). If you want to model the onset of the epidemic, it is better to include all of these factors in a single model and see which factors are significant. I understand that the covariates have different scales, but these can easily be accommodated using appropriate hierarchical modeling strategies (see Gelman & Hill's book on the topic if needed). Along similar lines, I can't figure out why the authors have chosen not to do their (rather routine) statistical analysis using a standard statistical package (R, SAS, Stata, etc). While it is entirely possible that the authors coded thing appropriately in Python it (1) seem like reinventing the wheel and (2) is impossible to tell whether their models are correct without going through all of the rather lengthy model code (which was provided, but seems like a totally unnecessary action given that using a standard package would automatically provide assurance that the model is correct). Furthermore, the authors have chosen to test significance using permutation tests. While this should be okay, it could (again) be done in a standard statistical package but seems inferior to just using the asymptotic estimates that would be provided via, for example, Cox regression. (The authors argue that their data set is small, but it seems like they should have 19 regions * 6 seasons of data regarding epidemic onset times which seems like it would be ample.)

I don't understand why 2 regions of France are excluded from the analysis (Limousin and Languedoc-Roussillon). I doubt that adding the two regions would significantly alter the results, but they should be included if at all possible (or at least reasons for their exclusion should be given).

How were the data for each region summarized? By this I mean, each region has a broad range of temperatures, population densities, humidities, and travel patterns and thus has to be summarized for use in the model. So, for example, when a regional temperature is summarized, is this done using a population density weighted mean? (You wouldn't want temperature from unpopulated areas being included.) In the case of mobility flows, do you take a distance weighted average? These details need to be provided in order to assess the validity of the modeling strategy.

It is unclear if random factors were used. The authors state in some places that random factors were used (e.g. lines 176, 179) and in other places that they were not used (e.g. lines 199, 271). It would seem that random factors should be used for the hierarchical modeling that should be occurring, but it is unclear. The authors need to be consistent in their statements.

The authors talk about "movement of sick individuals" in their mobility analyses (and their model reflects this concept). Sick individuals are likely to have different movement patterns than healthy individuals. For example, if someone is severely ill, they may be confined to a bed for the duration of their infectious period. Hopefully, ill individuals will choose to self-quarantine and not attend work or go to public places. Thus it seems unreasonable to model the onset of epidemic influenza based on movement of sick individuals. Really, one would want to model sub-clinical cases (i.e. individuals who have the disease but are still "healthy"), but there are no data for this. In the end, it probably doesn't matter for the outcome (assuming the subclinical cases = clinical cases * proportion), but it would be good state this assumption and/or change the language related to this issue. If the authors truly want to talk about the movement of sick individuals, the movement patterns should be altered to reflect likely changes in behavior.

Validity of the findings

Given the issues I have outlined with respect to the writing of the manuscript and the modeling that was done, it is unclear to me whether the findings are valid. (Particularly when it seems that "sub-models" were run each covariate without a model that unified all covariates and a "homemade" Python script was used for analyses.) Further clarity on how data were processed prior to use in the analyses is needed to accurately assess validity.

Additional comments

In their paper "Linking influenza epidemic onsets to covariates at different scales using survival analysis models" the authors use several spatially varying factors (temperature, humidity, mobility, and demography) to model when seasonal influenza epidemics start in France. The underlying idea of modeling onset times using a spatiotemporal model could potentially be of interest to public health agencies in France (and perhaps more broadly if others chose to employ the same modeling strategy). I have a number of concerns however regarding the presentation of the work and the modeling that was performed that should be addressed before the work would be publishable in PeerJ.

Annotated reviews are not available for download in order to protect the identity of reviewers who chose to remain anonymous.

---

## Round 0.3 · Minor Revisions

Your resubmitted manuscript has been re-reviewed by the original referees. As before, one review is "public" (i.e., non-anonymous) while the other is anonymous.

Both reviewers have additional suggestions. It is not my intention to have you submit revisions ad infinitum but I do think that these new concerns should be somehow addressed. A number of these points involve relatively small clarifications to notation.

Note that one of the referees has uploaded an annotated version of the manuscript.

Let me highlight a few points that the referees mentioned. However your next revision should address all the points raised by the referees (when I say address, I mean, address somehow --- this does not [necessarily] mean changing the paper on each and every point, but if you choose not to make a given change, an explanation should be provided).

Both referees question your permutation-based CIs. Please clarify and/or modify. In figure 2, some use of jitter or plotting symbol size should be used to discriminate overlapping points.

The estimates in table two should be clarified --- which parameters are these, in parameter notation?

Thank you.

Sincerely,
Andrew Noymer

·

Basic reporting

The article meets standards for publication for this criteria.

Experimental design

The model, data, and results remain poorly described:
- I continue to have major concerns with the ways that the methods are described - it is not clear to me what parameters are being estimated. For example, it is not clear to me what Table 2 is showing.
- In the second term for the equation for phi (line 463 of track-changed manuscript), the sum is taken over an index j which does not appear in the term being summed. This appears to be an error of notation.
- The notation for lambda on line 430 is not consistent with that used on line 619.
- Overlapping points in Figure 2 are not acceptable. At least jitter, or display in a different way.

Validity of the findings

While the revision addressed some of my major concerns from the first round, it raised additional fundamental concerns with the execution and presentation of the models.

- Bootstrap-based CIs would be a much more standard choice than permutation-based CIs. What is the theoretical justification for using the sampling distribution of the parameters under permutation?
- Paragraph starting on 638 states that non-epidemic years were left out. This is a somewhat justified approach, given that the outcome is only defined for epidemic years, but the limitations this introduces in terms of the interpretability of the results (e.g. results only being relevant for epidemic year dynamics) are substantial, and undiscussed.

Reviewer 2 ·

Basic reporting

The authors have greatly improved the clarity of the manuscript since the previous submission. That being said, in the attached PDF I have suggested numerous changes and potential areas where the manuscript could be shortened. There are some inconsistencies in notation in spots that certainly need to be corrected.

Experimental design

No comment.

Validity of the findings

I have some concerns about the identifiability of the model the authors propose. Given that

lambda = beta * phi ^alpha

and

beta = exp(X*b)

thus

lambda = exp(alpha*log(phi) + X*b)

the parameters end up being present in complex combinations (e.g., alpha and c seem particularly confounded). Therefore, I'm not sure given the current model specification, anyone would ever correctly infer these parameters (given perfect data). This could be why the authors had difficulty with the c parameter.

Additional comments

I appreciate the effort the authors have put into revising their previous submission in accordance with my concerns and those of the other reviewer. Most of my comments in the new version are related to improving clarity and quality of the writing. Other than those, I think the manuscript would be improved by including the results from the analysis with the 5 "missing" epidemics (perhaps as in the supplement). This way readers can evaluate for themselves whether dropping these data was a good choice. My only other substantive comment on this version is that the authors discard the idea of using asymptotic approximations for p-values in favor of permutation tests. See my comments in the attached PDF regarding this issue. The most important of these comments basically is that, if the authors are going to favor permutation tests so heavily, they should include some of the drawbacks of permutation tests (I listed a couple of these in my comments). I think it is also interesting to note that the asymptotic approximations agreed fairly well (in my opinion) with the permutations tests (with only 1 qualitative difference).

Annotated reviews are not available for download in order to protect the identity of reviewers who chose to remain anonymous.

---

## Round 0.4 · Minor Revisions

First of all, I would like to apologize for the delay in rendering this decision, and I thank you for your patience.

One of the referees feels that lines 221-239 in the tracked-changes version of the newest submission are a retrograde step as regards clarity of the model. This is the section of text that begins "To determine the scales at which epidemic..." and ends, "... onset week by week".

I have now given your paper a read, myself, and I would like to invite you to do two things. First, can you please clarify that your MATLAB program run_flucovariates.m is what corresponds to this section? Clarify this to me, not (necessarily) for the reader. If that is not right, what is the program that corresponds to this section of text?

Second, I invite you to edit this section once more for clarity. Since you provide code, I think a would-be reader who wants to do a "deep dive" on what you have done... can do so. I just want to verify (per above) that run_flucovariates.m is the right code. At the same time, in light of the referee comment, I would like to accord you the opportunity to edit this section to make it even clearer about what, precisely, you have done.

After that, I will accept your paper (without further peer review).

Thanks again for your patience.

·

Basic reporting

No comment.

Experimental design

I continue to have questions about how the model is formulated and don't think that the methods are described with sufficient detail to replicate the analysis. The new text (lines 221-239 in tracked changes document) does not clearly state what the linear mixed model regression uses as an outcome. Furthermore, the distributions of the random effects are not specified. With each iteration, the paper appears to be getting more confusing instead of having a clearer description of the methods.

Validity of the findings

No comment.

---

## Round 0.5 · accepted · Accept

I am satisfied that the manuscript is clear enough at this point, in conjunction with the code-made-available. I look forward to seeing the final-published version!